# Relative Importance of Deterministic and Stochastic Processes on Soil Microbial Community Assembly in Temperate Grasslands

**DOI:** 10.3390/microorganisms9091929

**Published:** 2021-09-10

**Authors:** Nana Liu, Huifeng Hu, Wenhong Ma, Ye Deng, Qinggang Wang, Ao Luo, Jiahui Meng, Xiaojuan Feng, Zhiheng Wang

**Affiliations:** 1Institute of Ecology and Key Laboratory for Earth Surface Processes of the Ministry of Education, College of Urban and Environmental Sciences, Peking University, Beijing 100871, China; nnliu@pku.edu.cn (N.L.); luoao@pku.edu.cn (A.L.); mengjiahui@pku.edu.cn (J.M.); 2State Key Laboratory of Vegetation and Environmental Change, Institute of Botany, Chinese Academy of Sciences, Beijing 100093, China; huifhu@ibcas.ac.cn; 3College of Ecology and Environment, Inner Mongolia University, Hohhot 010021, China; whmapku@126.com; 4CAS Key Laboratory of Environmental Biotechnology, Research Center for Eco-Environmental Sciences, Chinese Academy of Sciences, Beijing 100085, China; yedeng@rcees.ac.cn; 5Key Laboratory of Biodiversity and Organic Farming, Department of Ecology and Ecological Engineering, College of Resources and Environmental Sciences, Agricultural University, Beijing 100193, China; wangqg@cau.edu.cn

**Keywords:** community composition, environmental selection, dispersal limitation, subsoil microbial diversity, rare microbes, bacteria, archaea

## Abstract

Changes in species composition across communities, i.e., β-diversity, is a central focus of ecology. Compared to macroorganisms, the β-diversity of soil microbes and its drivers are less studied. Whether the determinants of soil microbial β-diversity are consistent between soil depths and between abundant and rare microorganisms remains controversial. Here, using the 16S-rRNA of soil bacteria and archaea sampled at different soil depths (0–10 and 30–50 cm) from 32 sites along an aridity gradient of 1500 km in the temperate grasslands in northern China, we compared the effects of deterministic and stochastic processes on the taxonomic and phylogenetic β-diversity of soil microbes. Using variation partitioning and null models, we found that the taxonomic β-diversity of the overall bacterial communities was more strongly determined by deterministic processes in both soil layers (the explanatory power of environmental distance in topsoil: 25.4%; subsoil: 47.4%), while their phylogenetic counterpart was more strongly determined by stochastic processes (the explanatory power of spatial distance in topsoil: 42.1; subsoil 24.7%). However, in terms of abundance, both the taxonomic and phylogenetic β-diversity of the abundant bacteria in both soil layers was more strongly determined by deterministic processes, while those of rare bacteria were more strongly determined by stochastic processes. In comparison with bacteria, both the taxonomic and phylogenetic β-diversity of the overall abundant and rare archaea were strongly determined by deterministic processes. Among the variables representing deterministic processes, contemporary and historical climate and aboveground vegetation dominated the microbial β-diversity of the overall and abundant microbes of both domains in topsoils, but soil geochemistry dominated in subsoils. This study presents a comprehensive understanding on the β-diversity of soil microbial communities in the temperate grasslands in northern China. Our findings highlight the importance of soil depth, phylogenetic turnover, and species abundance in the assembly processes of soil microbial communities.

## 1. Introduction

Changes in community species composition across space (i.e., β-diversity) is a central focus of ecology and is vital for understanding community assembly processes [1,2]. Although this issue has been widely explored in terrestrial plant and animal communities [3,4], it has been much less investigated in microbial communities [1,5,6]. Most studies on microbial β-diversity have been focused on topsoil (0–10 cm) [7,8], while few studies have examined microbial β-diversity in subsoils (below 30 cm) [9]. Compared to topsoils, subsoils generally have lower substrate availability, longer substrate turnover time in organic matter, and better insulation from the atmosphere [10], all of which may lead to the fact that subsoils are more sensitive and vulnerable to climate change [11,12]. Therefore, exploring the biogeographic patterns of β-diversity and the driving mechanisms in subsoil microbial communities will improve our understanding not only of the assembly processes of soil microbial communities but also of the responses of soil microbial communities to environmental changes.

Soil microbial β-diversity is frequently determined by two dominant processes: deterministic and stochastic processes [1,2]. Deterministic processes suggest that microbial β-diversity is determined by abiotic and biotic factors that represent the effects of environmental selection on microbial community composition [1]. In contrast, stochastic processes suggest that microbial β-diversity is determined by dispersal limitation, ecological drifts, genetic mutation, and historical contingencies [1,2]. Evaluating the relative contributions of deterministic and stochastic processes is critical for understanding the mechanisms of microbial community assembly but remains a challenge in current studies. Several statistical approaches have been developed to examine the relative importance of deterministic and stochastic processes, among which the distance–decay relationship (DDR) have been widely used [13]. This method evaluates the changes in the community composition along spatial and environmental gradients [14]. A steeper DDR slope implies a faster turnover in the species composition across a landscape [15]. Null models have also been used to infer ecological stochasticity by comparing the observed patterns with the expected random patterns produced by null models [6,16].

Studies have found that the relative importance of the deterministic and stochastic processes on soil microbial communities varies across environmental gradients (e g., pH, nutrients, and climate) [17,18], succession stages [19], and different functional assemblages [20,21]. However, the relative importance of these driving factors across different soil depths is unknown. Studies have shown that soil microbial communities are controlled by deterministic processes more than by stochastic processes in topsoils (0–10 cm) and that the deterministic factors are mainly climate-related factors [2]. In contrast, given the relatively well-insulated and oligotrophic environments in subsoils, the microbial communities in subsoils may also be strongly driven by deterministic processes but through different environmental factors, such as soil geochemical factors (i.e., soil pH) [17,22]. However, whether the drivers of microbial β-diversity are consistent across different soil depths remains to be tested.

Previous studies on terrestrial plants and animals have demonstrated that β-diversity and its drivers differed between abundant and rare microorganisms due to their different dispersal abilities [3] and environmental preferences [23,24]. Specifically, studies have suggested that the β-diversity of abundant plants and animals tends to be determined by deterministic processes [3,23,24], while that of rare microorganisms tends to be determined by stochastic processes [3]. Similar to plants and animals, most microbial communities normally contain large numbers of rare microbes and a few highly abundant microorganisms [25]. Current studies show inconsistent evidence for the relative importance of deterministic and stochastic processes on the β-diversity of rare and abundant soil microbes [25,26,27]. The generality of the previous findings based on plants and animals for soil microbial communities remains poorly known.

Here, we evaluated the microbial β-diversity and its drivers in top- (0–10 cm) versus subsoils (30–50 cm) along an aridity gradient of 1500 km in temperate grasslands in Inner Mongolia, China. The transect covers arid to mesic ecosystems and spans a wide range of climate and soil physicochemical conditions with varying plant species richness. In a previous study, we demonstrated the geographical patterns in soil bacterial and archaeal α-diversity along this transect and their drivers using 16S rRNA sequencing [28]. In the current study, we compared the relative effects of deterministic and stochastic processes on the β-diversity of soil bacterial and archaeal communities in both topsoils and subsoils. With these analyses, we aim to test the following hypotheses: (1) Soil microbial β-diversity in both soil layers is determined by deterministic processes, but the role of deterministic processes is stronger in subsoils than in topsoils; (2) the β-diversity and its drivers differ between abundant and rare soil microbes due to their different dispersal abilities and environmental preferences. Deterministic processes dominate the β-diversity of abundant microbes, while stochastic processes dominate that of rare microbes; (3) the environmental drivers representing the effects of deterministic processes on microbial β-diversity are distinct across soil depths and among taxa with different abundances.

## 2. Materials and Methods

### 2.1. Soil Samples and Data Collection

We collected 32 soil samples along an ~1500-km aridity transect from arid to mesic grasslands in Inner Mongolia (Longitude: from 107.929 to 119.970° E; latitude, from 39.154 to 49.618° N; Appendix A), China, in August 2015 [29]. The transect is in a temperate climatic zone with wind from Siberia all year round and varied climatic, edaphic, and vegetation conditions. It covers several vegetation types, including desert steppe (DS), typical steppe (TS), and meadow steppe (MS), from the southwest towards the northeast [30]. The desert steppe is arid, has low plant species richness, and is dominated by perennial drought–adaptive species including *Stipa klemenzii* and *Stipa breviflora*, etc. [30]. The typical steppe has the highest coverage in Inner Mongolian, with intermediate levels of net primary productivity (NPP) and plant species richness. The dominant species in the typical steppe are *S. grandis*, *S. krylovii*, and *Artemisia frigida*, etc. The meadow steppe has the highest NPP and plant species richness, and the dominant species are *Stipa baicalensis* and *Leymus chinensis*, etc. [30]. The mean annual precipitation (MAP) increases from 165.0 to 411.5 mm, and the mean annual temperature (MAT) decreases from 6.4 to −2.3 °C from the southwest towards the northeast. Soil types along this transect include Calcisols, Kastanozems, and Calcic Chernozems [30]. The soil pH and soil organic carbon content in the topsoils along this transect range from 7.7 to 9.9 and from 0.39 to 4.69%, respectively (see Appendix A).

Soil samples were collected from two relatively contrasting soil depths: 0–10 cm (topsoil) and 30–50 cm (subsoil) [28]. A total of seventeen environmental variables related to climate, plants, and soil properties were compiled or measured (see Appendix A). The climate-related variables included MAP, MAT, aridity index, soil water content (SWC, mm month^−1^), and historical temperature anomaly. The historical temperature anomaly was calculated as the contemporary MAT minus that during the Last Glacial Maximum (LGM, i.e., the most recent glaciation, ca. 21,000 to 18,000 years before present day) [31]. Contemporary MAP and MAT data from 1950 to 2000 and data for the MAT during the LGM were obtained from the WorldClim website (http://worldclim.org/version2 accessed on 10 May 2017) [32], and the aridity index and SWC data were obtained from https://cgiarcsi.community/ accessed on 10 May 2017.

The plant-related variables included aboveground vegetation biomass, plant species richness, and NPP. The vegetation biomass and plant species richness were measured during sampling, and the NPP was obtained from the Numerical Terradynamic Simulation Group (NTSG) (http://www.ntsg.umt.edu/project/modis/mod17.php; Missoula, MT, USA accessed on 10 May 2017). The soil-related variables including soil total nitrogen, soil total carbon, soil organic carbon, soil total phosphorus, soil pH, soil extractable Fe, soil extractable Al, soil clay content, soil silt content, and soil sand content were measured and used (see Appendix A). More details related to soil sampling and soil parameter measurement were described in [28].

### 2.2. DNA Extraction, High-Throughput Amplicon Sequencing and Sequence Processing

Total soil DNA was extracted using the MoBio PowerSoil DNA isolation kit (MoBio Laboratories, Carlsbad, CA, USA). The V4 region of the 16S rRNA gene for bacteria was amplified using the barcoding primer pair 515F/806R (515F, 5′-GTGCCAGCMGCCGCGGTAA-3′; 806R, 5′-GGACTACHVGGGTWTCTAAT-3′) [33]. A region of the 16S rRNA gene for archaea was amplified using the primer pair 1106F/1378R (1106F, 5′-TTWAGTCAGGCAACGAGC-3′; 1378R, 5′-TGTGCAAGGAGCAGGGAC-3′) [34]. The DNA samples were sent to Novogene (Beijing, China) for sequencing using an Illumina HiSeq2500 platform.

Sequences generated from amplicon-based sequencing were processed using UPARSE (Tiburon, CA, USA) [35] on the Galaxy pipeline in the Metagenomics for Environmental Microbiology (http://mem.rcees.ac.cn:8080) [36] in 15 June 2017. Finally, a total of 3,531,946 and 4,086,723 high-quality sequences were obtained for the soil bacteria and archaea and were grouped into 23,458 and 3152 OTUs, respectively, at a 97% sequence similarity cut-off. The sequence data in each community were resampled to 32,885 and 50,347 sequences per sample (the smallest number of sequences per sample across the samples) for bacteria and archaea, respectively. The 16S rRNA sequence data were analyzed in a previous study [28], and further analyses were conducted in this study.

We defined ‘abundant’ microbes as OTUs with an abundance ≥ 0.1% in all samples, and the ‘rare’ microbes were defined as OTUs with an abundance ≤ 0.01% in all samples. These thresholds were modified from Dai et al. [37].

### 2.3. Statistical Analysis

#### 2.3.1. Estimation of Soil Microbial β-Diversity

The pairwise Bray–Curtis similarity (taxonomic β-diversity) and weighted UniFrac similarity (phylogenetic β-diversity) between each pair of communities were calculated using the ‘vegan’ and ‘phyloseq’ packages [38]. β-diversity patterns were visualized by nonmetric multidimensional scaling ordination (NMDS) with significant groupings at the 95% confidence interval. There were three different nonparametric multivariate statistical tests (Adonis, permutational multivariate analysis of variance; ANOSIM, analysis of similarity; MRPP, multiresponse permutation procedure) that were used to test the significance of β-diversity variation between the two communities. Multivariate homogeneity of group dispersions (variances) was conducted to test the multivariate dispersion between the topsoils and subsoils at *p* < 0.05 [39]. NMDS, nonparametric multivariate statistical tests, and multivariate dispersion were performed using the ‘vegan’ package.

#### 2.3.2. The Relationship between Microbial β-Diversity and Distance

Spatial distance was calculated from the geographic coordinates (latitude and longitude) of the sampled sites using the ‘geosphere’ package. The composite environmental distance between each pair of communities was calculated as the Euclidean distance generated from a normalized combination of six environmental groups, including (1) historical temperature anomaly; (2) contemporary climate; (3) aboveground vegetation; (4) soil fertility; (5) soil pH; and (6) soil mineral content (Appendix A). The relationships between the microbial β-diversity and both the spatial and environmental distances were analyzed using ordinary least-squares regressions to demonstrate the distance–decay curves. The slopes of these distance–decay curves were extracted to represent the species turnover rate. The significance of the distance–decay curves was tested using the Mantel test [40], and the significance of the difference in the slopes of the distance–decay curves between the two regression lines was assessed by analysis of variance (ANOVA) and pairwise comparisons of least squares means (LSMeans) using the ‘lsmeans’ package [41].

#### 2.3.3. Variation Partitioning and Null Model Analysis

Variation partitioning was conducted to determine the relative contribution of spatial versus environmental distances to microbial β-diversity using the ‘vegan’ package. Using this method, the total variance in the microbial β-diversity was partitioned into four parts: the independent and shared effects of spatial and environmental distances and residuals [42]. Null model analyses (999 randomizations) were used to decipher community assembly mechanisms and to provide critical insights into the role of variable selection, homogenous selection, homogenous dispersal, dispersal limitation, and drift in shaping microbial β-diversity [6]. The evaluation of the five community assembly processes was based on the phylogenetic β-nearest taxon index (βNTI) and Raup–Crick (RC_Bray_) β-diversity metrics [6]. The βNTI was based on a null model test of the phylogenetic β-diversity index βMNTD (β mean nearest-taxon distance), and RC_Bray_ (modified Raup-Crick index) was based on a null model test of the Bray–Curtis taxonomic β-diversity index. A significant βNTI (i.e., |βNTI| > 2) indicates the dominance of the selection effect on microbial community assembly. More specifically, βNTI > 2 indicates significantly more phylogenetic turnover than expected (i.e., variable selection), while βNTI < −2 indicates significantly less phylogenetic turnover than expected (i.e., homogenous selection). |βNTI| < 2 indicates the dominance of stochasticity, which, in combination with RCBray, further deciphers the effect of stochasticity. Specifically, |βNTI| < 2 and RC_Bray_ < −0.95 indicate homogenous dispersal. |βNTI| < 2 and RC_Bray_ > 0.95 indicate dispersal limitation. |βNTI| < 2 and |RC_Bray_| < 0.95 indicate ‘undominated processes’, including weak selection, weak dispersal, and diversification and drift [1,6].

#### 2.3.4. Random Forest Analysis

Random forest analysis [43] was conducted to determine the importance of spatial distance and individual environmental factors in structuring microbial β-diversity. The importance of each factor was determined by evaluating the increase in the mean square error between observations and out-of-bag predictions when the factors were randomly permuted in over 500 trees [44]. The merit of random forest analysis is to alleviate multicollinearity problems in multivariate analysis by building bagged tree ensembles and by including a random subset of features for each tree (500 trees). The random forest analysis was conducted using the ‘randomForest’ package.

#### 2.3.5. Estimation of Habitat Niche Breadth

The habitat niche breadth of a microbial community is an important trait underlying the relative importance of deterministic and stochastic processes [45]. We calculated the community habitat niche breadths (Levins’ niche breadths) using the ‘niche.width’ function in the ‘spaa’ package in R [46]. The significance of the difference in the habitat niche breadths between the topsoils versus the subsoils and among the three communities was evaluated by paired t-tests and Tukey’s multiple comparisons at *p* < 0.05 using the ‘HSD.test’ function in the ‘agricolae’ package.

All of the analyses were conducted in R software (version 3.6.2; Vienna, Austria).

### 2.4. Accession Numbers

HiSeq2500 sequencing data have been deposited in the public National Center for Biotechnology Information (NCBI) database with the accession number PRJNA557316.

## 3. Results

### 3.1. Microbial Communities in Top- and Subsoils

All three nonparametric multivariate statistical tests (Adonis, ANOSIM, and MRPP) for the nonmetric multidimensional scaling ordination analysis indicated that the taxonomic and phylogenetic β-diversity for bacteria and archaea significantly differed between the top- and subsoils (*p* < 0.05) (Figure 1a–d and Appendix A). Specifically, the taxonomic β-diversity for the archaea was significantly more dispersed in the subsoils than in the topsoils (*p* < 0.05, Figure 1e,f). The topsoils generally contained more *Acidobacteria*, *Alphaproteobacteria*, *Verrucomicrobia*, *Planctomycetes*, *Bacteroidetes*, *Betaprotobacteria*, *Firmicutes*, and *Crenarchaeota* than subsoils based on their relative abundance, while the subsoils contained more *Actinobacteria*, *Nitrospirae*, *Gemmatimonadetes*, *Euryarchaeota*, and *Parvarchaeota* (Figure 1g,h). In addition, the bacterial and archaeal communities in both soil layers included a large number of rare microbial taxa with low abundances and few abundant taxa with high abundances (Figure 1i,j).

### 3.2. Distance–Decay Relationships in Top- and Subsoils

As indicated by the distance–decay curves, the taxonomic and phylogenetic similarity between the microbial communities significantly decreased with both spatial and environmental distances in both the bacterial and archaeal communities (*p* < 0.05; Figure 2a,b,d,e,g–k), indicating distance–decay patterns. The slopes of the distance–decay curves indicate the community turnover rate, and we found that the bacterial communities in the topsoils had faster turnover rates in the taxonomic and phylogenetic β-diversity than those in the subsoils (Figure 2c,f). In contrast, archaeal communities had inconsistent community turnover rates in the taxonomic and phylogenetic β-diversity in both the topsoils and subsoils (Figure 2i,l). The slopes of the distance–decay curves across spatial and environmental distances indicate the relative importance of spatial and environmental distances, and we found that the phylogenetic β-diversity for the bacterial communities in the topsoils was more strongly associated with spatial distance, while the taxonomic β-diversity in the subsoils was more strongly associated with environmental distance (Figure 2c,f). In comparison, the archaeal β-diversity in both soil layers was more strongly associated with environmental distance (Figure 2i,l).

We further found that the taxonomic and phylogenetic β-diversity of the abundant and rare microorganisms for the bacterial and archaeal communities showed significant (*p* < 0.05) distance–decay patterns with both spatial and environmental distances in both soil layers (Figure 3a–d,f–i,k–n,p–s), with the exception of the phylogenetic β-diversity of the rare bacterial and archaeal taxa in subsoils. Interestingly, the β-diversity of abundant bacterial taxa was more strongly associated with environmental distance, while rare bacterial taxa were more strongly associated with spatial distance, as illustrated by the comparison of the slopes along the environmental versus the spatial distances (Figure 3e,j). In comparison, the β-diversity of both the abundant and rare archaeal taxa was consistently associated with environmental distance in both soil layers (Figure 3o,t).

### 3.3. Relative Importance of Spatial versus Environmental Distances on Microbial β-Diversity

Together, the environmental and spatial distances explained the higher variations in the β-diversity (taxonomic and phylogenetic: 53.3 and 37.0%, respectively) of the overall bacterial communities in the topsoils but the lower variations in the subsoils (only 27.5 and 15.3%, respectively) (Figure 4a). The taxonomic β-diversity of the overall bacterial communities in both soil layers was mostly determined by deterministic processes, as indicated by the higher unique effect of environmental distance (topsoil, 25.4%; subsoil, 47.4%) than spatial distance (topsoil, 22.9%; subsoil, 7.5%; Figure 4b). Environmental distance showed a higher unique effect on the taxonomic β-diversity of the overall bacterial communities in the subsoils than in the topsoils when standardizing the unique effect (Figure 4b). However, the phylogenetic β-diversity of the overall bacterial communities in both soil layers was mostly determined by stochastic processes, as indicated by the higher unique effect of the spatial distance (topsoils, 42.1%; subsoils, 24.7%) than the environmental distance (topsoils, 9.9%; subsoils, 24.3%; Figure 4b).

Similarly, the spatial and environmental distances together explained the higher variation in the β-diversity of the abundant (taxonomic and phylogenetic: 30.1 and 28.1%, respectively) and rare (53.7 and 10.7%, respectively) bacterial microorganisms in the topsoils but the lower variation in the β-diversity in the subsoils (15.9 and 16.5% for abundant; 16.1 and 5.8% for rare; Figure 4a). The abundant bacterial microbes were mostly determined by deterministic processes in both soil layers, as indicated by the higher unique effect of the environmental distance (taxonomic and phylogenetic in topsoils: 38.9 and 38.7%; 33.3 and 54.7% in subsoils) than the spatial (12.0 and 12.0% in topsoils; 16.7 and 4.1% in subsoils) distance on both β-diversities (Figure 4b). Rare bacterial microbes were mostly determined by stochastic processes in both soil layers, as indicated by the higher unique effect of the spatial distance (taxonomic and phylogenetic in topsoils: 35.4 and 72.2%; 59.1 and 90.6% in subsoils) than the environmental (14.4 and 0.01% in topsoils; 2.5 and 9.3%in subsoils) distance on both β-diversities (Figure 4b).

In comparison with the bacterial communities, together, the spatial and environmental distances explained the higher variation in the β-diversity of the overall (taxonomic and phylogenetic: 53.7 and 29.0%), abundant (43.5 and 36.9%), and rare (30.3 and 4.7%) archaeal taxa in the topsoils, but relatively lower variation in the β-diversity in the subsoils (overall: 44.8 and 29.6%; abundant: 44.0 and 24.9%; rare: 28.1 and 0.5%; Figure 4c). Overall, the abundant and rare archaeal taxa were consistently determined by deterministic processes, as indicated by the higher unique effect of the environmental distance than the spatial distance on the β-diversity (Figure 4d). Environmental distances showed a higher unique effect on the β-diversity of the archaeal communities in the subsoils than in the topsoils (Figure 4d).

### 3.4. Ecological Mechanisms Underlying Microbial β-diversity

To uncover the ecological mechanisms driving microbial β-diversity, the phylogenetic β-nearest taxon index (βNTI) and Bray–Curtis-based Raup–Crick (RC_Bray_) metrics were calculated based on null model analysis (Figure 5). Across all sites, the bacterial communities in the topsoils were mostly determined by stochastic processes, especially ‘undominated processes’, such as weak selection, weak dispersal, diversification, and drift. However, the bacterial communities in the subsoils were equally determined by deterministic and stochastic processes (Figure 5a,c). Rare bacterial microbes (such as *Thalassomonas*, *Sinorhizobium*, *Solirubrobacter*, etc.) in both soil layers were mostly determined by stochastic processes (e.g., ‘undominated processes’). By comparison, the archaeal communities and rare archaeal microbes in both soil layers were mostly determined by deterministic processes (e.g., homogenous selection; Figure 5b,d).

### 3.5. Environmental Drivers of Microbial Community Composition

Using random forest analyses (Figure 6), we found that among the spatial and the environmental distances of the six individual environmental variables, historical temperature anomaly was the main driver of both β-diversity measures (taxonomic and phylogenetic) of the overall bacterial microbes in topsoils, while the soil pH was the main driver in the subsoils. By comparison, aboveground vegetation was the main driver of both β-diversity measures of the abundant bacterial microbes in both soil layers (Figure 6). Historical temperature anomaly, contemporary climate, and spatial distance were the top three drivers of both β-diversity measures of the rare bacterial microbes in the topsoils, while the spatial distance and the historical temperature anomaly were the top two drivers in the subsoils (Figure 6).

In comparison with bacteria, contemporary climate, and aboveground vegetation were the top two drivers of both β-diversity measures of the overall, abundant, and rare archaeal taxa in the topsoils, while the soil pH and contemporary climate were the top two drivers of the β-diversity for the overall, abundant, and rare archaeal taxa in the subsoils (Figure 6).

### 3.6. Niche Breadth

The habitat niche breadths in the communities of the overall, abundant, and rare bacterial microbes were broader than those in the corresponding archaeal taxa in both soil layers (Figure 7). The communities of the abundant microbes of the two microbial domains had the widest niche breadths followed by those of the overall taxa, and the communities of rare microbes had the narrowest niche breadths (*p* < 0.05; Figure 7). More importantly, the habitat niche breadths were significantly broader (*p* < 0.05) in the subsoils than in the topsoils for all three bacterial communities but were significantly narrower in the subsoils than in the topsoils for all three archaeal communities (Figure 7).

## 4. Discussion

### 4.1. Consistent Community Assembly Processes across Soil Depths

We observed a strong distance decay of community similarity along both the spatial and environmental distances in the bacterial and archaeal communities at both soil depths, which suggests that bacterial and archaeal β-diversity is jointly determined by stochastic and deterministic processes. This finding is consistent with the continuum hypothesis, suggesting that stochastic and deterministic processes interact to structure the assembly of microbial communities [20,21]. However, we found that the slopes of the distance–decay curves for bacterial taxonomic β-diversity and archaeal β-diversity (both taxonomic and phylogenetic) were higher across the environmental distance than across the spatial distance (Figure 2). Analyses of variation partitioning also showed that the environmental distance explained the higher variation in the bacterial taxonomic β-diversity and archaeal β-diversity (both taxonomic and phylogenetic) than the spatial distance (Figure 4). These results suggest that both bacterial taxonomic β-diversity and archaeal β-diversity in both soil layers tend to be determined by deterministic processes, supporting our first hypothesis. This is consistent with previous studies on agricultural ecosystems [22], grasslands [8,18], and aquatic ecosystems [5], all of which showed stronger effects of deterministic processes, especially in terms of homogenous selection to structure microbial β-diversity.

Deterministic processes tend to occur when long-distance dispersal is not limited [47]. Long-distance dispersal is possible in the grasslands of Inner Mongolia because there are winds and dust storms all year round that can transport near-surface soil hundreds of miles away. Previous studies found that wind speed or dust transport could influence the microbial distribution in water and in the near-surface atmosphere and make their geographic patterns more responsive to climatic and soil factors [48,49]. In addition to long-distance dispersal, strong environmental heterogeneity, such as the aridity gradient and vegetation types in harsh arid drylands may also lead to the strong effect of environmental factors on soil microbial composition changes, even across large spatial scales.

Though soil depth did not influence the identity of community assembly processes, it changed the relative strength of these processes. Specifically, in comparison with the topsoils, deterministic processes had a higher effect in the subsoils for both the bacterial and archaeal communities in terms of taxonomic β-diversity (Figure 4), suggesting that subsoil microbial communities are more strongly responsive to changing environment (such as soil pH, discussed in the last section) than those in topsoils (multiple environmental factors, historical climate, contemporary climate, and aboveground vegetation). The higher deterministic effects in subsoils than in topsoils are consistent with previous studies on prokaryote and fungal communities in drylands [11,12]. However, the low but still significant effect of deterministic processes in topsoil are not well consistent the above-mentioned studies [11,12]; however, they are consistent with the bacterial communities in the topsoils of Tibetan grasslands [9]. This indicates that the topsoil microbial communities might be barely determined solely by the assembly mechanism due to complex geography and environmental conditions across landscapes, while the subsoil microbial communities might be easily deciphered based on deterministic mechanisms due to similar environmental conditions (lower substrate availability, better insulation from the atmosphere) across landscapes. The contrasting strength of assembly processes demonstrates the important role of soil–horizon partitioning in filtering specific microbial taxa in topsoils (such as *Acidobacteria*, *Proteobacteria*, *Planctomycetes*, *Bacteroidetes*, *Firmicutes*) and subsoils (such as *Actinobacteria*, *Nitrospirae*, *Gemmatimonadetes*) [9]. In the deterministic processes, heterogeneous and variable environmental conditions in topsoils, especially in near-surface soils support the co-occurrence of numerous microbial taxa, while the enclosed and stable environment in subsoils only supports the occurrence of oligotrophic taxa and high pH-adapted taxa.

Most studies exploring community assembly mechanisms based on variation partitioning rarely compared the discrepancy between different β-diversity metrics (taxonomic versus phylogenetic) [17,22]. However, we found that the taxonomic and phylogenetic β-diversity of bacterial communities was determined by different processes based on variation partitioning. In contrast to taxonomic β-diversity, phylogenetic β-diversity was mainly driven by dispersal limitation in both soil layers, which does not support our first hypothesis. This finding is in contrast with a recent finding that bacterial communities are mainly determined by stochastic processes in topsoils and by deterministic processes in subsoils [12]. We suspect that phylogenetic β-diversity, including both weighted UniFrac and βMNTD, considers phylogenetic relatedness between OTUs, while taxonomic β-diversity only considers OTU abundance [38]. In other words, phylogenetic β-diversity includes additional information on the evolution of bacterial OTUs compared to taxonomic β-diversity [50]. These results suggest that stochastic processes may play an important role in phylogenetic turnover among soil bacterial communities, which provides insights that are not accessible via the taxonomic turnover method [51]. More studies are needed to further decipher the assembly mechanisms of microbial communities using taxonomic versus phylogenetic β-diversity.

### 4.2. Distinct Community Assembly Processes for Abundant and Rare Bacterial Microbes

Both abundant and rare bacterial microbes showed a decrease in community similarity across spatial and environmental distances, showing distance–decay patterns. However, abundant bacterial microbes had a higher taxonomic and phylogenetic turnover (higher slopes of distance-decay curves) than rare bacterial microbes in both soil layers (Figure 3). This is in contrast to our expectation but is consistent with previous findings in forest ecosystems [52] and oil-contaminated soil [53]. One possible reason for the different species turnover (slopes in DDRs) between abundant and rare bacterial microbes may be because abundant microbes are more highly aggregated than rare microbes, which influences the distance decay in community similarity along both the spatial and environmental distances [52].

Despite the similarities in the β-diversity of abundant and rare bacterial microbes, we found that abundant bacterial microbes were more strongly determined by deterministic processes, while rare bacterial microbes were more strongly determined by stochastic processes, which supports our second hypothesis. In general, abundant microbes (such as *Rubrobacter*, *Bradyrhizobium*, *Mycobacterium*, and *Steroidobacter*) were all cosmopolitan (100% across sites in our studies), had high local abundance, and occupied a wider niche (Figure 1 and Figure 6). Thus, they could competitively use different kinds of resources and could adapt to different environments (such as the response to aboveground vegetation, discussed in the last section) more effectively [53]. Therefore, abundant microbes may be less limited by dispersal than rare microbes, even at large geographic scales and thus can sufficiently be delivered to suitable habitats with suitable environmental conditions [47]. In contrast, rare microbes (such as *Nocardioides*, *Actinomadura*, *Inquilinus*, and *Paenibacillus*) normally have low local abundance and weak competitive ability. Therefore, the β-diversity of rare bacterial microbes is more strongly driven by stochastic processes, including weak environmental filtering and weak dispersal.

The finding on the distinct community assembly mechanisms between abundant and rare bacterial microbes provides insights into the inconsistent argument on the assembly mechanisms between abundant and rare microbial taxa in numerous ecosystems (such as grasslands, agriculture, and water ecosystems) [25,26,27]. Our study demonstrated that abundant microbes tended to be determined by environmental filtering, perhaps due to their good adaptation to the surrounding environment and strong dispersal ability, while rare microbes tended to be determined by stochasticity due to the effects of their weak dispersal ability and historical contingency on their distributions [3].

### 4.3. Distinct Environmental Drivers of β-Diversity across Soil Depths and Microbial Groups

Although deterministic processes predominated the bacterial and archaeal β-diversity in both soil layers, the effect of the environmental factors representing the effects of deterministic processes varied across soil depths, confirming our third hypothesis. In general, the bacterial and archaeal β-diversity in the topsoils tended to be determined by contemporary and past climate or aboveground vegetation, while the β-diversity in the subsoils tended to be determined by edaphic factors, e.g., soil pH. These results suggest that different environmental conditions across soil depths may have filtered different microbial community compositions.

In topsoils, bacterial β-diversity was dominated by historical temperature anomaly rather than by contemporary climates. Similarly, recent studies have found that the paleoclimate since the Last Glacial Maximum (LGM) explains more variation in the composition of bacterial communities than the current climate does because the paleoclimate may influence the compositions of soil bacterial communities by its effect on present-day microbial enzyme sensitivity or soil properties [54,55]. In contrast, the archaeal β-diversity in the topsoils was dominated by analyzing the contemporary climate and aboveground vegetation rather than by means of the historical temperature anomaly, which suggests that archaeal communities are more rapidly responsive to recent climate dynamics and plant carbon input compared to bacteria [56]. For example, the archaeal *Parvarchaeota*, an acidophilic group [57], can assimilate multiple carbon resources (e.g., starch, cellulose, disaccharides) and is positively responsive to aboveground plant biomass [28].

In contrast to topsoils, microbial taxa in subsoils, such as *Actinobacteria*, are more adaptive to oligotrophic environments and are more related to soil geochemistry [10]; hence, the microbial β-diversity in subsoils tends to be determined by edaphic factors rather than by climate and aboveground vegetation. Indeed, our results demonstrated that soil pH was a stronger environmental driver in structuring microbial communities in subsoils than in topsoils in Inner Mongolia grasslands, which is in contrast to previous studies on other ecosystems showing a stronger pH effect on the soil microbial communities in topsoils than in subsoils [17].

## 5. Conclusions

In the present study, we found that in general, deterministic processes had stronger effects on soil bacterial and archaeal β-diversity in both top- and subsoils than stochastic processes. However, we found that the relative effects of deterministic and stochastic processes on β-diversity and the major environmental drivers representing deterministic process varied between taxonomic and phylogenetic β-diversity, between the different soil layers, and between the abundant and rare microbes. The taxonomic β-diversity of the overall bacterial communities and the β-diversity of the abundant bacteria was more strongly determined by deterministic processes, while the phylogenetic β-diversity of the overall bacterial communities and the β-diversity of rare bacteria was more strongly determined by stochastic processes. In contrast, archaeal β-diversity was consistently more strongly determined by deterministic processes. The microbial β-diversity in the topsoils and for the abundant microbes was more strongly determined by contemporary and historical climate and aboveground vegetation, while the microbial β-diversity in the subsoils was more strongly determined by soil geochemistry. This reminds us to comprehensively consider multiple aspects of microbial communities when evaluating their assembly processes, which would improve our understanding of the assembly of soil microbial communities under a changing environment.

## Figures and Tables

**Figure 1 microorganisms-09-01929-f001:**
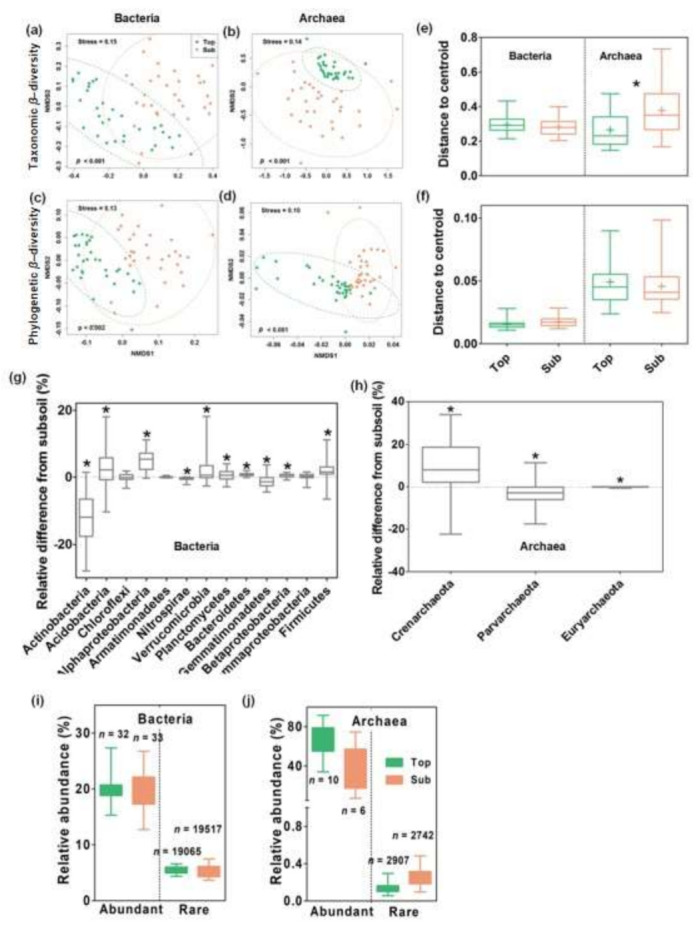
Soil bacterial and archaeal community composition across soil depths in temperate grasslands. Taxonomic and phylogenetic β-diversity was evaluated using Bray–Curtis and weighted UniFrac distances, respectively. (**a**–**d**) Nonmetric multidimensional scaling (NMDS) ordination showed differences in the taxonomic and phylogenetic β-diversity of the bacterial and archaeal communities between topsoils (green) and subsoils (salmon). Dashed lines represent significant groupings in both soil layers at the 95% confidence interval, and the difference between soil layers was statistically tested by all three nonparametric multivariate statistical tests (Adonis, ANOSIM and MRPP; Appendix A). (**e,f**) Analysis of multivariate homogeneity of group dispersions (variances) was conducted to compare the taxonomic or phylogenetic β-dispersion between the topsoils and subsoils. (**g**,**h**) The relative difference in abundance between the topsoils and subsoils for different microbial groups was calculated as the relative abundance of groups in topsoils minus that in subsoils. The asterisk (*) represents significant difference of the relative abundance between top- and subsoils at *p* < 0.05. (**i**,**j**) Panels indicate the relative abundance of abundant and rare microorganisms across soil depths, with *n* indicating the OTU numbers.

**Figure 2 microorganisms-09-01929-f002:**
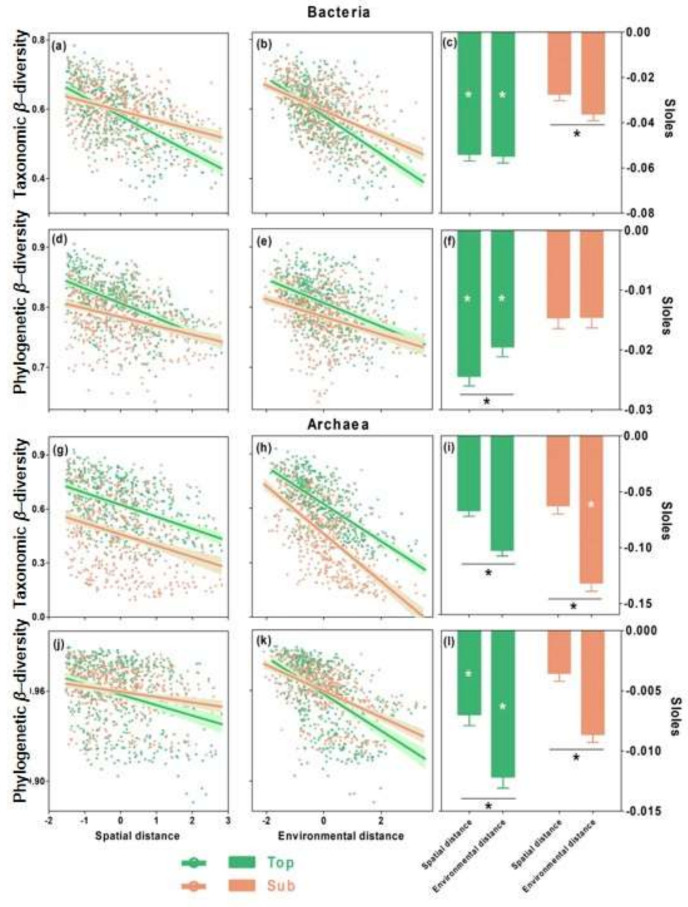
Changes in taxonomic and phylogenetic β-diversity of bacterial (**a**–**f**) and archaeal (**g**–**l**) communities as functions of spatial and environmental distances and their corresponding slopes in topsoils and subsoils. Taxonomic and phylogenetic β-diversity was evaluated using Bray–Curtis and weighted UniFrac similarities, respectively. Green and salmon colours indicate topsoils and subsoils, respectively. Solid lines represent significant linear regressions (*p* < 0.05) for the relationships between the β-diversity and the distances evaluated by the Mantel test. The slopes between the β-diversity and the spatial/environmental distances were compared between the top- and subsoils (* on the bars) and between the spatial and environmental distances (* below the bars) at *p* < 0.05. Environmental distance was generated by computing the Euclidean distance among six environmental variables, including historical temperature anomaly, contemporary climate, aboveground vegetation, soil fertility, soil pH, and soil mineral content (Appendix A). Spatial and environmental distances were standardized to be comparable.

**Figure 3 microorganisms-09-01929-f003:**
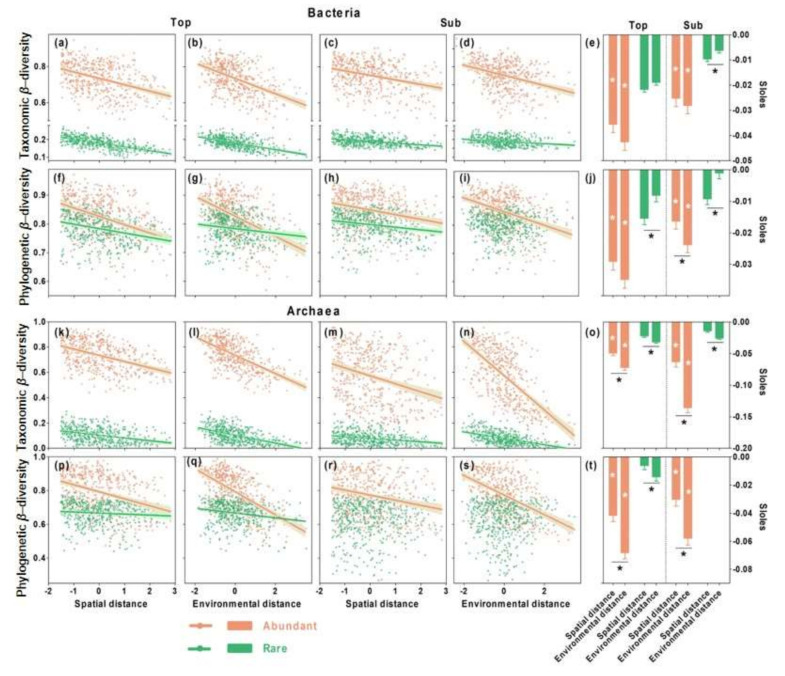
Taxonomic and phylogenetic β-diversity of abundant and rare bacteria (**a**–**j**) and archaea (**k**–**t**) taxa as functions of spatial and environmental distances and their corresponding slopes in topsoils and subsoils. Taxonomic and phylogenetic β-diversity was evaluated using Bray–Curtis and weighted UniFrac similarities, respectively. Green and salmon colours indicate abundant and rare microorganisms, respectively. Solid lines represent the significant linear regressions (*p* < 0.05) for the relationships between the β-diversity of the abundant (sample) and rare (green) taxa and distances evaluated by the Mantel test. The slopes between the β-diversity and spatial/environmental distances were compared between the abundant and rare microorganisms (* on the bars) and between the spatial and environmental distances (* below the bars) at *p* < 0.05. Environmental distance was estimated as the Euclidean distance in the environmental space of the six environmental variables, including historical temperature anomaly, contemporary climate, aboveground vegetation, soil fertility, soil pH, and soil mineral content (Appendix A). Spatial and environmental distances were standardized to be comparable.

**Figure 4 microorganisms-09-01929-f004:**
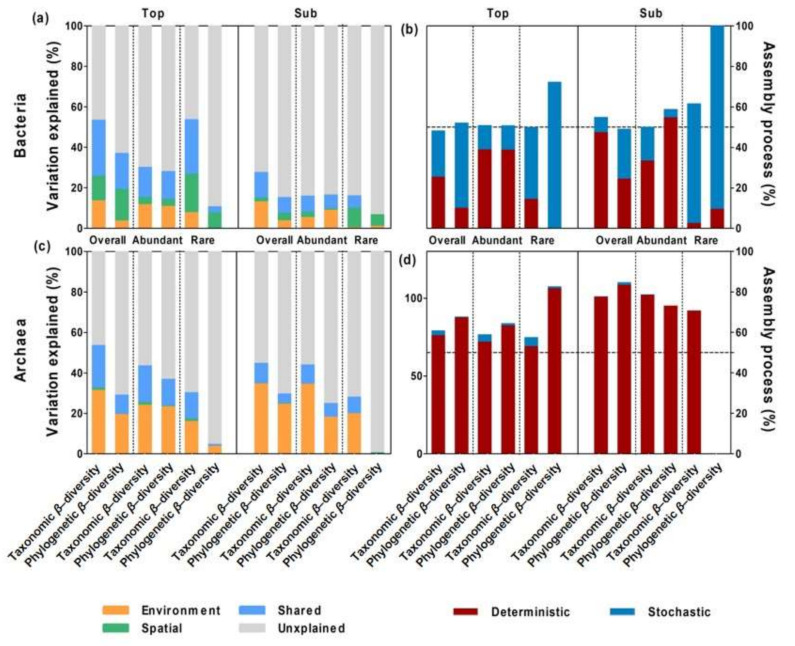
Relative contribution of environmental and spatial distances and of deterministic and stochastic processes to the β-diversity of bacterial (**a**,**b**) and archaeal (**c**,**d**) communities in topsoils and subsoils. Taxonomic and phylogenetic β-diversity was evaluated using Bray–Curtis and weighted UniFrac similarities, respectively. The effects of spatial and environmental distances on microbial β-diversity were evaluated using partial redundancy analysis, which partitioned the total variance into four parts: the independent spatial effect (green), independent environmental effect (salmon), shared (light blue) effects of the spatial and environmental distances, and residuals (grey). The deterministic processes (%, red) were calculated as the percentage of the effects of environmental distance on the total effect of the environmental and spatial distances, while the stochastic processes (%, dark blue) were calculated as the percentage of the effect of the spatial distance on the total effect of the environmental and spatial distances.

**Figure 5 microorganisms-09-01929-f005:**
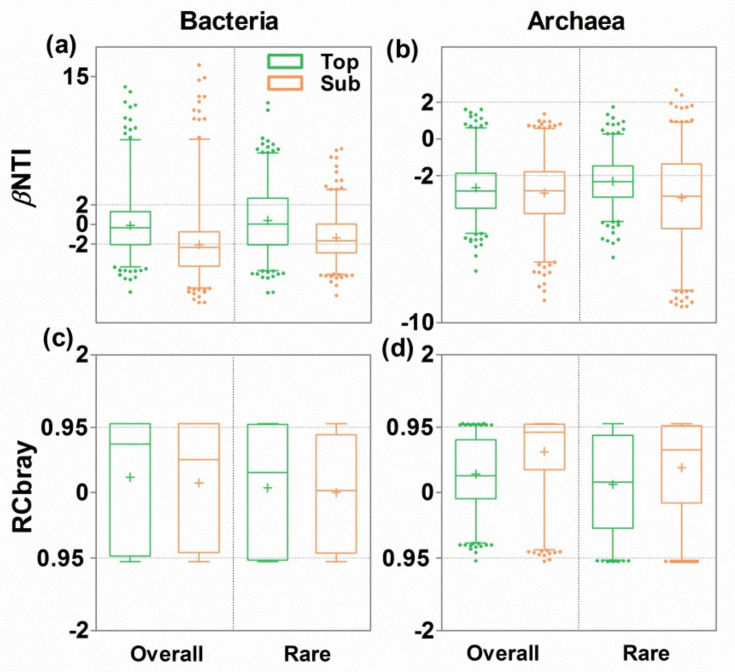
Variation in the *β*-nearest taxon index (βNTI, (**a**,**b**)) and Bray–Curtis-based Raup–Crick matrix (RC_Bray_, (**c**,**d**)) based on null model expectations for bacterial (**a**,**c**) and archaeal (**b**,**d**) communities in topsoils and subsoils. “+” in each box indicates mean values. βNTI > 2 indicates variable selection, and βNTI < −2 indicates homogenous selection. |βNTI| < 2 and RC_Bray_ < −0.95 indicate homogenous dispersal. |βNTI| < 2 and RC_Bray_ > 0.95 indicate dispersal limitation. |βNTI| < 2 and |RC_Bray_| < 0.95 mainly indicate ‘undominated processes’, such as weal selection, weak dispersal, diversification, and drift. Due to the cosmopolitan properties (100% across sites in our studies) of the abundant microbes, the βNTI of the abundant microbes could not be calculated if the conspecific taxa in different communities were not excluded from MNTD calculations using the “comdistnt” functions in the “picante” packages of R.

**Figure 6 microorganisms-09-01929-f006:**
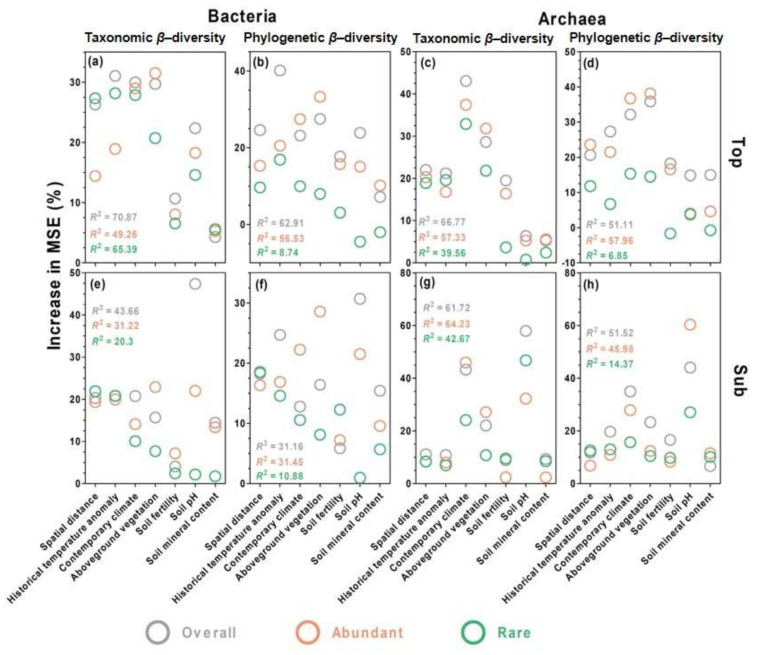
Relative contribution of the spatial distance and the distance of each individual environmental variable to soil bacterial (**a**,**b**,**e**,**f**) and archaeal (**c**,**d**,**g**,**h**) β-diversity in topsoils and subsoils. Taxonomic (**a**–**d**) and phylogenetic (**e**–**h**) β-diversity was evaluated using Bray–Curtis and weighted UniFrac similarities, respectively. The results were generated from random forest analyses.

**Figure 7 microorganisms-09-01929-f007:**
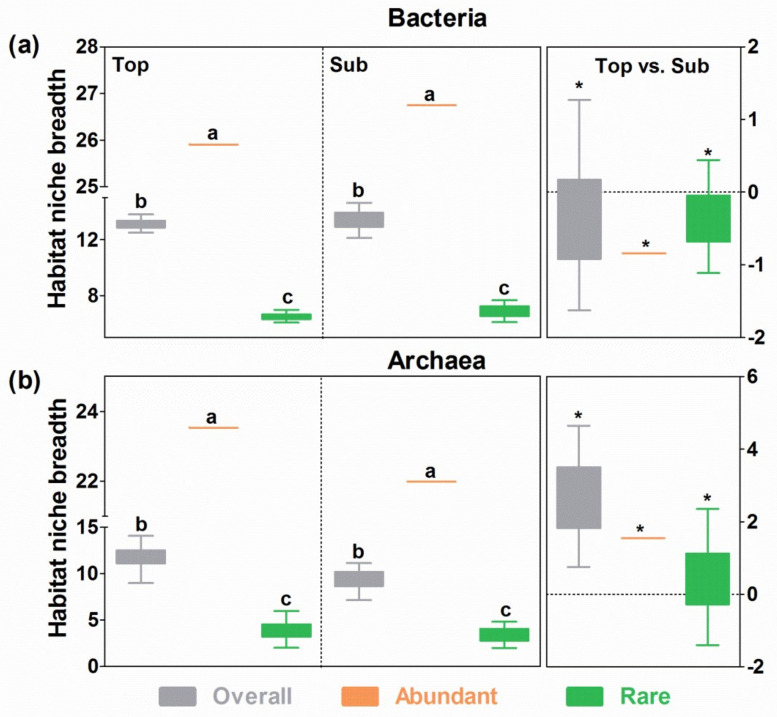
Habitat niche breadths of bacterial (**a**) and archaeal (**b**) communities in topsoils and subsoils. The right panels indicate the difference (*, *p* < 0.05) in habitat niche breadths between topsoils and subsoils. The letters above bars indicate significance (*p* < 0.05) among overall, abundant, and rare microbial communities for bacteria and archaea, while the asterisks above bars on the right indicate significant differences (*p* < 0.05) between topsoils and subsoils for all microbial communities.

## Data Availability

The names of the accession number(s) can be found in the paper.

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
