# Peer review of "Relative Importance of Deterministic and Stochastic Processes on Soil Microbial Community Assembly in Temperate Grasslands"

_microorganisms, 2021, doi:10.3390/microorganisms9091929_

Round 1

Reviewer 1 Report

Title: Relative importance of deterministic and stochastic processes on soil microbial community assembly in temperate grasslands
Abstract: The abstract is well written. It encapsulates the entire study (a bit of introduction, aim, result and outcome). However, if there the section could be shortened a bit to make it more concise, it would be great. 
Introduction:
I find this section well written as it gives a good background of the research in question. Also the aim of the study is evident in the beginning and concluding parts. 
Line 51 – 52: I agree that subsoils are more affected by climate change factors such as warming and there are numerous publications confirming this fact. Could you please make some citations to that effect?
Materials and Methods: 
I find this section well-structured and scientifically valid. I will only suggest some description of the study area is provided. 
Results: The results are well presented. 
Discussion: This section is fairly good but will require more justifications with some more comparisons with similar studies in the past.
The manuscript is well concluded as the main outcomes are well captured with some recommendations
Although, the study is interesting and will appeal to readers, I suggest the comments raised (from all reviewers) are duly addressed to make it more comprehensible and concise to bring the quality to a publishable level. Also grammar revision should be done

Reviewer 2 Report

This paper presents a comprehensive understanding of the β-diversity of soil microbial communities in temperate grasslands in northern China. I think it is a useful paper because the authors' findings highlight the importance of soil depth, phylogenetic turnover, and species abundance in the assembly processes of soil microbial communities. The revised version of this article is well written, well structured, and clear. The title clearly describes the contents of the paper. The abstract provides a concise and complete summary and the reference list is appropriate. Although the language is not bad, I still feel that editing the text by a native English speaker would help and improve the readability of the paper. The authors should also carefully check the paper for misprints, e.g. line 38, change: norther China to northern China.

However, this is an interesting paper, I would recommend the publication of the manuscript. 
